# Drosophila immune priming to *Enterococcus faecalis* relies on immune tolerance rather than resistance

Kevin Cabrera[1,2], Duncan S. Hoard[1], Olivia Gibson[3], Daniel I. Martinez[1], Zeba Wunderlich[2,3]*

1 Department of Developmental and Cell Biology, University of California, Irvine, California, United States of America, 2 Biological Design Center, Boston University, Boston, Massachusetts, United States of America, 3 Department of Biology, Boston University, Boston, Massachusetts, United States of America

* zeba@bu.edu

## Abstract

Innate immune priming increases an organism's survival of a second infection after an initial, non-lethal infection. We used *Drosophila melanogaster* and an insect-derived strain of *Enterococcus faecalis* to study transcriptional control of priming. In contrast to other pathogens, the enhanced survival in primed animals does not correlate with decreased *E. faecalis* load. Further analysis shows that primed organisms tolerate, rather than resist infection. Using RNA-seq of immune tissues, we found many genes were upregulated in only primed flies, suggesting a distinct transcriptional program in response to initial and secondary infections. In contrast, few genes continuously express throughout the experiment or more efficiently re-activate upon reinfection. Priming experiments in immune deficient mutants revealed Imd is largely dispensable for responding to a single infection but needed to fully prime. Together, this indicates the fly's innate immune response is plastic—differing in immune strategy, transcriptional program, and pathway use depending on infection history.

## Author summary

Most animals, like the fruit fly *Drosophila melanogaster*, lack an adaptive immune system and react to infection using only an innate immune response. In this paper, we study how previous infection with the bacterium *Enterococcus faecalis* changes the immune response to a second infection with the same bacterium, through a phenomenon called immune priming. We find that primed flies tend to survive more, tolerate a higher bacterial load, and undergo priming-specific gene expression reprogramming compared to non-primed flies. We also find that eliminating a key component of the Imd pathway, which is not canonically related to response to *E. faecalis* lowered priming ability in flies. These experiments highlight the true complexity of fly immune response and provide a basis for further exploring the interrelatedness of multiple known innate immune pathways in regulating a complex phenomenon like immune priming.

**Data Availability Statement:** The RNA-seq data in the paper is deposited on GEO using accession GSE210282. The remainder of the data is available as supplemental tables and in a GitHub repository:

https://github.com/WunderlichLab/
ImmunePriming-RNAseq.

**Funding:** This work was funded by National
Science Foundation grant MCB-1953312/2223888
(to Z.W.) K.C. is a National Institutes of Health-
Initiative for Maximizing Student Development
(GM055246) Fellow and a recipient of a National
Science Foundation Graduate Research
Fellowships Program award. The funders had no
role in study design, data collection and analysis,
decision to publish, or preparation of the
manuscript. https://www.nsf.gov https://nih.gov.

**Competing interests:** The authors have declared
that no competing interests exist.

## Introduction

The fruit fly *Drosophila melanogaster* inhabits environments rich in bacteria, fungi, and viruses. The fly has to mitigate these pathogens to survive. To this end, it has evolved a tightly controlled innate immune response. It has long been appreciated that the fly immune pathways can distinguish between Gram-positive bacteria and fungi versus Gram-negative bacteria [1]. Recent findings have elaborated on these models by showing specificity within Gram-classifications, cross-talk between the two individual pathways, and a coordination between tissues [2–4].

Among these refined characteristics is the potential for immune memory in the innate immune system. While flies lack the canonical antibody-mediated immune memory of the adaptive immune response, an initial non-lethal infection can sometimes promote survival of a subsequent infection. This phenomenon, termed immune priming, has been observed in evolutionarily distant organisms such as plants [5], multiple arthropod species [6], and mammals [7,8]. The fact that this mechanism is present in animals that have an adaptive response hints at its importance in organismal fitness.

Despite immune priming's effect on survival, the underlying mechanism controlling it in flies is not completely understood. Three hypotheses have been proposed to explain the physiological effects of priming [9,10]. The first is that there is a qualitatively different response, e.g. a difference in the identity of the effectors produced or cellular processes, between primed versus non-primed insects, leading to a more effective response. A second hypothesis is that insects will initiate an immune response during priming, but will re-initiate the same immune function in a potentiated manner upon reinfection. This is most similar to the phenomenon of what has been observed in mammalian trained immunity [8]. Lastly, immune effectors created during the initial immune response may be persistently expressed, eliminating the lag time in initiating effector production. Since flies can harbor low-level chronic infections instead of completely clearing them [11,12], these chronic infections may contribute to immune priming by providing a consistent mild stimulus. Priming may be driven by a combination of these three mechanisms. Delineating the relative contributions of each may not only reveal the drivers of infection survival, but may also suggest epigenetic mechanisms of gene regulation and tradeoffs between the immune response and other biological processes.

*Drosophila* is a good model for dissecting the mechanisms driving immune priming due to its genetic tractability, extensively characterized innate immune pathways, and its homology to mammalian innate immune pathways. There has been extensive characterization of the fly's transcriptional response to a variety of bacteria [13–15] and the progression of bacterial load during infection with different bacteria or in different host genotypes [11]. Studies of priming have revealed the key role of phagocytosis. Blocking phagocytosis in adults decreases priming with the Gram-positive bacterium *Streptococcus pneumoniae* [16]. Blocking developmental phagocytosis of apoptotic debris also makes larvae more susceptible to bacterial infection [17]. In addition, the production of reactive oxygen species as a result of wounding contributes to immune priming with the Gram-positive bacterium *Enterococcus faecalis* [18]. These findings lay the foundation for testing the mechanistic hypotheses that underlie immune priming.

In this study, we present a multifaceted approach to understand immune priming in the fly using an *E. faecalis* reinfection model. *E. faecalis*, a Gram-positive, naturally occurring pathogen of the fly, has been previously used to induce an immune response with dose-dependent lethality. We characterize not only the physiological response to priming by way of survival and bacterial load to immune priming, but also the transcriptional response that underlies the physiology. By assaying transcription separately in both the hemocytes and fat body, we explore the organ-specific program that mounts a more effective primed immune response.

## Results

### *E. faecalis* priming increases survival after re-infection

To determine whether we could elicit a priming response in flies, we needed to find appropriate priming and lethal doses. For these experiments, 4-day old male Oregon-R (OreR) flies were infected with a strain of the Gram-positive bacteria *Enterococcus faecalis* originally isolated from wild-caught *D. melanogaster* (Fig 1A) [19]. Survival was scored as the hazard ratio (HR) of the bacterial-infected flies against a PBS-injected control; a HR > 1 indicates worse survival of the experimental sample compared to the control. The HR also gives a quantitative summary of the survival curve–the higher the HR, the more quickly the animals died. Initial infection with *E. faecalis* showed dose-dependent survival (Fig 1B; *Efae* Low Dose vs. PBS

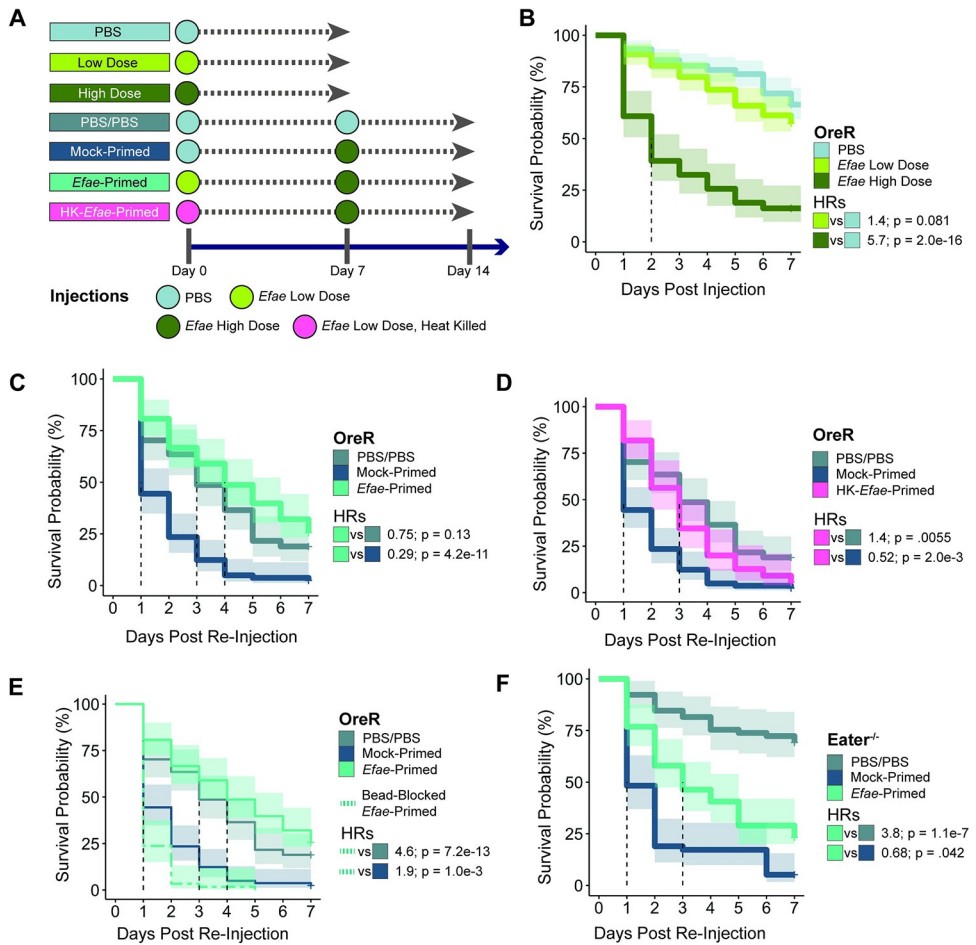

**Fig 1. *E. faecalis* can induce immune priming in *D. melanogaster*.** A). Schematic of single and double-injection experiments. B). Survival of OreR flies injected with PBS (n = 149), Efae Low Dose (~3,000 CFU/fly, n = 129), and *Efae* High Dose (~30,000 CFU/fly, n = 74). Dotted line indicates median survival time. Shaded area indicates 95% confidence interval. Pairwise hazard ratios (HR) are calculated using a Cox proportional hazard model, and Wald statistic p-values testing if the HR significantly differs from 1 are reported. P-values are Benjamini-Hochberg corrected for multiple testing. Full survival statistics can be found in S1 Table. C). Survival of primed OreR flies versus double-injected, non-primed controls. D). Survival of OreR flies primed with heat-killed *E. faecalis* (HK-*Efae*-Primed: n = 55) versus flies primed with live *E. faecalis*; data for *Efae*-Primed & Mock-Primed same as C. E). There is a complete loss of priming ability when phagocytes are bead-blocked in the initial low-dose *E. faecalis* infection. All data except for Bead-Blocked-*Efae*-Primed are the same as C, replotted for comparison. F). Survival of primed, phagocytosis-deficient *eater*-mutant flies versus double-injected, non-primed controls (PBS/PBS: n = 65, Mock-Primed: n = 58, *Efae*-Primed: n = 69).

HR = 1.4 [95% CI 0.96–2.1], *Efae* High Dose vs PBS HR = 5.7 [3.9–8.3]). Flies infected with a dose of ~30,000 CFU/fly (*Efae* High Dose) gradually died off, with more than fifty percent of flies dying by day 2, making it a practical choice for representing a lethal dose. Flies injected with a lower dose of ~3,000 CFU/fly (*Efae* Low Dose) had survival comparable to those injected with PBS, with a HR not significantly different from 1 (p = 0.081), indicating that death was largely due to the injection process itself, rather than from bacterial challenge.

To model re-infection, flies were initially injected either with a low bacterial dose (i.e. *Efae*-primed flies) or a negative control of PBS (i.e. Mock-primed flies) (Fig 1A). After resting for seven days, flies were re-injected with a high dose of *E. faecalis* and assayed. Seven days was chosen as the priming interval because we found that flies had gained enhanced re-infection survival from priming (S1A Fig), reached a stable chronic bacterial load (Fig 2A), and survived

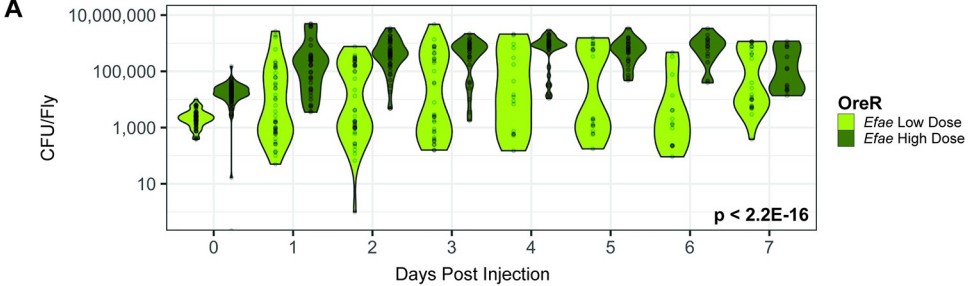

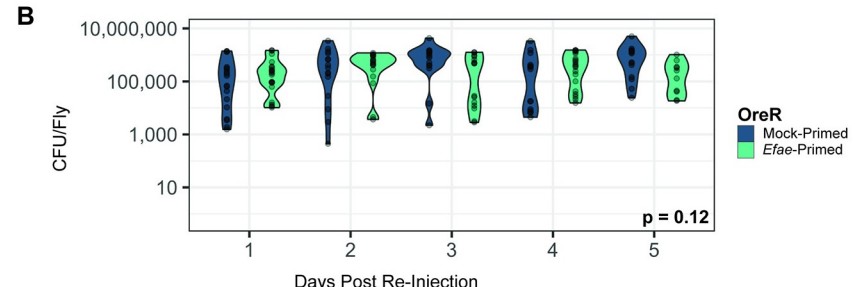

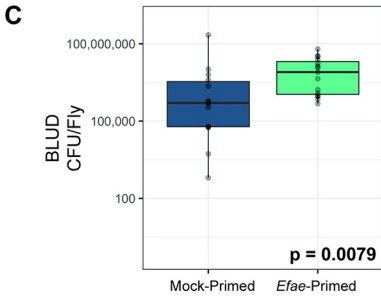

**Fig 2. Bacterial clearance is not correlated with primed survival against *E. faecalis* re-infection.** A). Bacterial load of single-injected flies. Flies were abdominally injected with either *E. faecalis* Low Dose (~3,000 CFU/fly) or *E. faecalis* High Dose (~30,000 CFU/fly), and a subset was dilution plated every 24 hours. There is a significant difference in bacterial load over time between initially low-dose and high-dose infected flies (Kruskal-Wallis rank sum test: df = 7, $X^2$ = 106.38, p < 2.2E-16). B). Bacterial load of double-injected flies. Mock-Primed and *Efae*-Primed flies do not differ in their bacterial load over time (Kruskal-Wallis rank sum test: df = 4, $X^2$ = 7.2423, p = 0.12). Data displays up to day 5 because of the strong survivor bias inherent to selecting flies that are still alive after that point (reference survival at day 5 and after in Fig 1C). C). Bacterial load upon death (BLUD) of double-injected flies (Wilcoxon rank sum test: W = 45, p = 0.0079).

in high enough numbers to practically collect for re-infection. We define priming as an increase in survival in *Efae*-primed flies compared to Mock-primed flies. Quantitatively, we assessed priming by comparing *Efae*-primed to Mock-primed survival using the HR; priming is indicated by a HR that is significantly less than 1 (*Efae*-Primed vs. Mock-Primed HR = 0.29, p = 4.2e-11, Fig 1C). Though there was a decrease in survival from double sterile wounding compared to a single sterile wound with PBS (S1B Fig), *Efae*-primed flies had survival comparable to this double-PBS injected baseline. We can again use the HR to define this "full" priming–when the *Efae*-primed flies survive as well as the double-PBS control, this results in a HR that is not different from 1 (*Efae*-Primed vs. PBS/PBS HR = 0.75, p = 0.13). In fact, *Efae*-primed flies not only survived as well as the PBS/PBS control, but also showed improved survival when compared to single, High Dose-infected flies (S1C Fig).

To see what bacterial signals are required for priming, we attempted to prime flies with heat-killed *E. faecalis*, which retains its signaling-responsive components but lacks any additional virulence factors [20,21]. This experiment resulted in a more moderate increase in survival rate compared to live bacteria priming (Fig 1D, HK-*Efae*-Primed vs. Mock-Primed HR = 0.52, p = 2.0e-3). As can be seen by comparing the HK-*Efae*-Primed survival curve to the PBS/PBS survival curve (HR = 1.4, p = 0.055), these animals do not achieve "full" priming as is the case with live bacteria. This implies some level of priming is conferred simply through bacterial sensing, but that the effect is not as robust as when the fly is exposed to the live microbe. This may either be because the live microbe produces other virulence factors or damage that is needed for priming or because the heat-killed microbe's products are cleared too quickly to create an equally strong priming response.

To compare *E. faecalis* priming to the priming described for *Streptococcus pneumoniae*, which was dependent on phagocytosis [16], we disrupted phagocytosis in two ways. We first blocked phagocytosis with beads during the initial *E. faecalis* infection in OreR flies as was done previously [16]. This caused a complete loss of priming ability (Fig 1E). An orthogonal method of assessing the role of phagocytosis in priming was done using an *eater* mutant [22]. The hemocytes in these flies are unable to carry out bacterial phagocytosis and have cell adhesion defects in the larva but can still mount a full Toll and Imd immune response [23]. By comparing the *Efae*-primed to Mock-primed flies, we can observe a modest amount of immune priming (HR = 0.68, p = 0.42) (Fig 1F). However, the *Efae*-primed flies died more quickly than the PBS/PBS controls. The HR comparing *eater Efae*-primed flies to the PBS/PBS control is greater than 1 (HR = 3.8, p = 1.1e-7, S1 Table), indicating that the mutants are unable to achieve full priming. Despite a difference in genetic background compared to the OreR bead blocking experiment, loss of phagocytosis still caused a loss in priming ability. Together, this indicates that phagocytosis is needed to fully prime.

## Priming increases tolerance of *E. faecalis*

To measure the infection dynamics underlying both the un-primed and primed response to *E. faecalis*, we tracked bacterial load throughout the course of the infection. Infected flies were collected at 24 hour intervals after injection, homogenized, and plated in a serial dilution. As a baseline, we followed bacterial load in flies solely injected with either a high (~30,000 CFU/fly) or low dose (~3,000 CFU/fly) of *E. faecalis* (Fig 2A). By day 2 after injection, the bacterial loads in flies infected with a high dose were generally above 100,000 CFU/fly. This indicates that without priming, the bacterial load in flies infected with a lethal dose increases to a high plateau. In contrast, by day 1 the distribution of bacterial loads in flies initially infected with a low dose was bimodal, consistent with what has been previously reported [11]. This suggests a subset of flies were more effectively resisting the infection and attempting to clear it, while another

subset tolerated a relatively high bacterial load. The data from the low dose flies indicate two things. First, even a low dose of *E. faecalis* is not completely eliminated from the animals. Second, upon reinfection, there are likely two distinct populations of flies, harboring either a relatively high or low bacterial burden, which could alter their capability to survive a subsequent infection.

We then tested the relationship between bacterial burden and the enhanced survival seen in primed flies. Flies that are primed could increase their survival by either more efficiently clearing the infection or more effectively tolerating a chronic bacterial burden. When looking at bacterial load in double-injected flies, there was no significant difference between Mock-primed and *Efae*-primed cohorts across the time course (Kruskal-Wallis rank sum test: p = 0.12) (Fig 2B). Despite their significant differences in survival (Fig 1C), this does not correlate with a difference in the bacterial load between the two conditions, indicating that the improved survival of *Efae*-primed flies relative to the Mock-primed flies is likely due to tolerance, not resistance. To further confirm that bacterial tolerance is driving the survival of *Efae*-primed flies, we also measured the bacterial load upon death (BLUD) [11] for double-injected flies. The higher an animal's BLUD, the higher its tolerance for a particular microbe. We found that *Efae*-primed flies harbored a significantly higher bacterial burden at the time of death (Fig 2C). This experiment further supports the idea that primed flies are able to tolerate a higher bacterial load than Mock-primed flies before they succumb to an infection.

## Fat bodies show priming-specific transcription

To correlate increased survival in primed flies with transcriptional response, we measured gene expression in the fat body using RNA-seq. The fly fat body is a liver-like tissue responsible for driving an extensive transcriptional program in response to bacterial infections [24,25]. As in previous experiments, flies were injected either singly or twice, with samples collected 24 hours after each injection (Fig 3A; S2 Table). To identify genes differentially expressed in response to each injection, we performed differential gene expression analysis against a non-injected, age-matched control. In this way, we generated four lists of up-regulated genes to compare–those upregulated in the animals with a single low dose infection, a single high dose infection, a mock-priming protocol, or a *Efae*-priming protocol. Genes that were differentially up-regulated only, for example, in *Efae*-primed flies were identified as "priming-specific". As a comparison to prior work, we analyzed the expression profiles of a previously published list of "core" immune genes in our samples and found a subset was induced upon infection in our samples (S2A Fig) [13].

The comparison of fat body transcription across conditions showed a high amount of *Efae* primed-specific and Mock-primed specific upregulation (149 genes & 408 genes, respectively, using an FDR cutoff of 0.05) (Fig 3B and 3C, full list for all conditions and overlap in S3 Table). Only a small fraction of these genes has been previously annotated with immune functions (19 *Efae*-Primed genes, ~13%; 15 Mock-Primed genes, ~4%) [13,26], although gene ontology (GO) analysis indicated immune response as one of the highest enriched terms (S2B Fig, top). Mock-primed specific GO term enrichment indicated response to stimuli, but also included genes involved specifically in response to mechanical stimuli and post-transcriptional gene regulation (S2B Fig bottom & S3 Table).

To delineate pathways whose component genes were upregulated in *Efae*-primed fat body versus Mock-primed fat body transcriptomes, we applied gene set enrichment analysis (GSEA) on the full transcriptome for both conditions. GSEA is an approach that looks for the coordinated up- or down-regulation of a set of genes involved in a common pathway or function. Since it uses all the transcriptome data, as opposed to differentially expressed genes

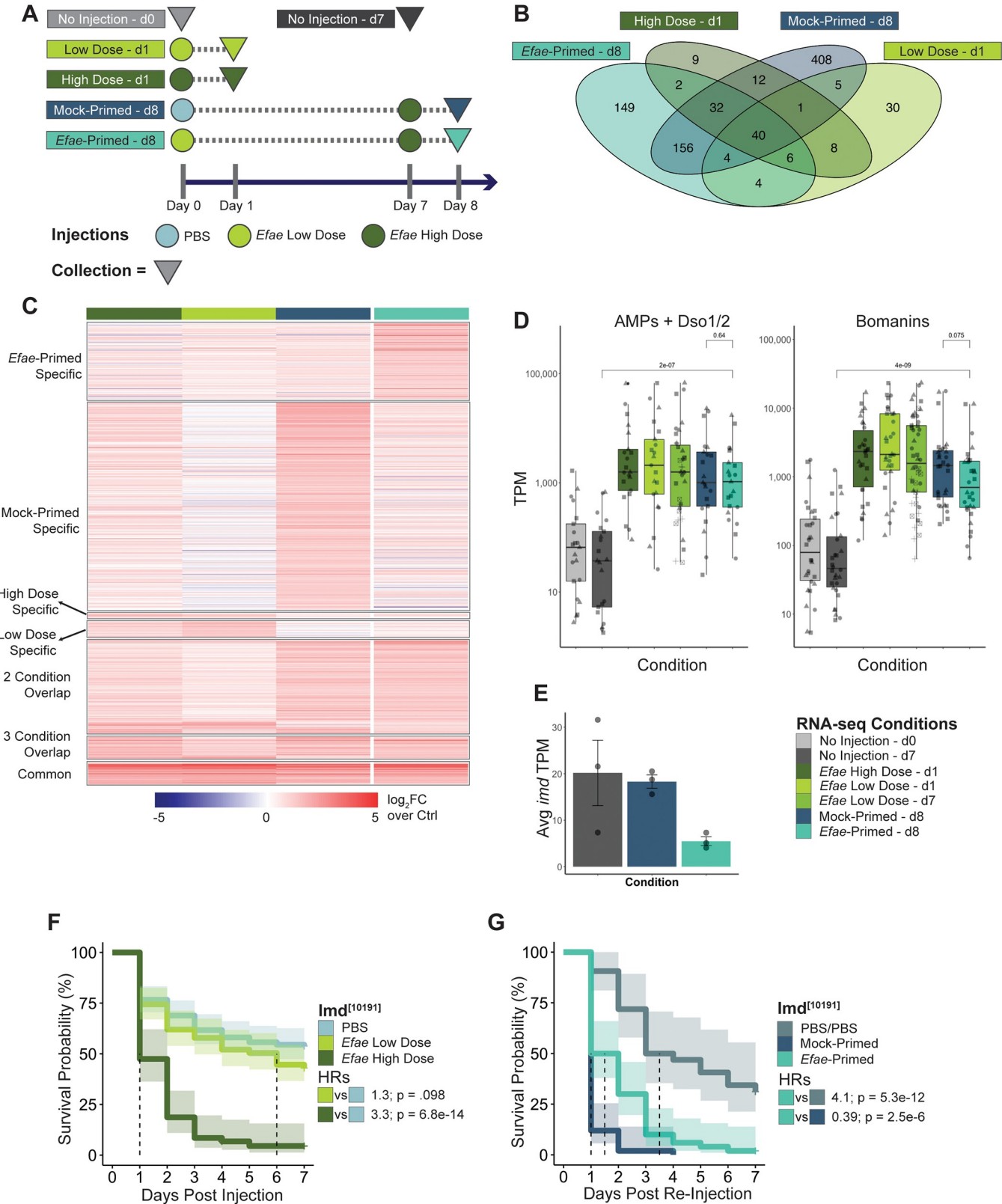

**Fig 3. Fat bodies have a high degree of priming-specific transcriptional up-regulation.** A). Sample collection for RNA-seq experiments. Conditions are the same as Fig 1A, with the addition of age-matched, non-injected controls at Day 0 and Day 7. Circles represent injections and triangles represent time of

collection. B). Venn-diagram of significantly up-regulated genes (log fold change (log$_2$FC) >1 & false discovery rate (FDR) <0.05) for conditions in A compared to age-matched controls. C). Heat map of significantly up-regulated genes as corresponding to B (scale: log$_2$FC over age-matched controls) D). Expression in log$_{10}$(TPM+1) of ubiquitously up-regulated AMPs [left] and *Bomanins* [right]. Biological replicates are designated by the shape of individual points. While there is a significant difference in AMP and *Bom* expression in *Efae*-Primed fat bodies compared to their age-matched, non-injected controls (Wilcoxon test; AMPs: p = 2.0E-07, *Boms*: p = 4.0E-09), there is not a significant difference in ubiquitous AMP and *Bom* expression between Mock-Primed and *Efae*-Primed fat bodies (Wilcoxon test; AMPs: p = 0.64, *Boms*: p = 0.075). P-values were corrected for multiple testing using the Bonferroni correction. E). Average TPMs for the gene *imd* in double-injected fat body samples. F). Survival of *imd*-mutant flies injected with PBS (n = 167), *Efae* Low Dose (n = 121), and *Efae* High Dose (n = 86). Dotted line indicates median survival time. Shaded area indicates 95% confidence interval. Pairwise comparisons are calculated using a Cox proportional hazard model with hazard ratios (HR), and Wald statistic values reported for experimental conditions versus their PBS negative control. G). Survival of primed *imd*-mutant flies versus double-injected, non-primed controls (PBS/PBS: n = 61, Mock-Primed: n = 60, *Efae*-Primed: n = 71).

identified by a fixed threshold, it can reveal differentially expressed pathways between samples that GO analysis may not detect [27]. Visualizing our GSEA results as a network of enriched terms identifies global enrichment trends in our dataset, rather than focusing on individual terms. *Efae*-primed samples were enriched for pathways involved in protein and lipid metabolism and metabolite transport, while Mock-primed fat bodies were enriched for pathways involved in the cell cycle (S3 Fig; full analysis in S4 Table). This suggests there is metabolic reprogramming associated with priming and altered regulation of cell division in Mock-primed fat bodies. Despite the high degree of unique transcriptional activity in Mock-primed fat bodies, Mock-primed flies die more quickly than either *Efae*-primed or high dose-infected flies. This suggests that this transcriptional reaction is not necessarily advantageous for infection survival. Taken together, fat bodies showed a strong transcriptional response to infection, with a high degree of Mock-primed and *Efae*-primed-specific transcription.

We also noted that all conditions shared a set of 40 commonly up-regulated genes, which we call "core genes." Seventeen of these core genes are known or suspected AMPs, including several *Bomanins* (*Boms*), *Daisho 1 & 2*, and the AMPs *Metchnikowin*, *Drosomycin*, *Diptericin B*, and *Baramicin A* (S2B Fig) [4,28–30]. Previous experimental work has shown that survival of *E. faecalis* infection is strongly dependent on the *Bom* gene family [31]. Flies lacking 10 out of the 12 *Boms* succumb to a single *E. faecalis* infection as quickly as flies that lack Toll signaling. Bacterial load data indicates that flies lacking either these 10 *Boms* resist an individual *E. faecalis* infection more weakly than wild type flies. Conversely, flies with deletions of several AMPs (4 *Attacins*, 2 *Diptericins*, *Drosocin*, *Drosomycin*, *Metchnikowin*, and *Defensin*) or *Baramicin A* show only modest decreases in survival of *E. faecalis* infections [4,29].

Given their differing effects on *E. faecalis* infection survival, we decided to analyze the expression patterns of the core *Boms* separately from the other core known or suspected AMPs. We displayed the distribution of expression levels of each gene group using transcripts per million (TPMs). When comparing expression of the core *Boms*, we found no significant difference in expression between the Mock-primed and *Efae*-primed flies (Wilcoxon rank sum test: p = 0.075) (Fig 3D, right). Likewise, a comparison of expression levels for the core AMP or AMP-like genes yielded no significant difference between the Mock-primed and *Efae*-primed flies (Wilcoxon rank sum test: p = 0.64) (Fig 3D, left). This indicates that increased survival of *Efae*-primed flies is not due to the primed fat bodies producing more transcripts associated with bacterial resistance. This observation is consistent with the lack of increased bacterial clearance for *Efae*-primed relative to Mock-primed flies in Fig 2B and further supports the notion that priming promotes survival through bacterial tolerance.

## Loss of Imd negatively impacts the fly's ability to prime against *E. faecalis*

We also observed priming-specific down-regulation of *imd* (Fig 3E), which led us to consider the role of Imd signaling in the priming response. While Imd signaling is canonically associated with response to Gram-negative bacterial infections, it is also connected to regulation of

the MAPK-mediated reactive oxygen species production and wound response, as well as a generalized stress response [32,33]. We first hypothesized that the downregulation of *imd* in *Efae*-primed flies might lead to lower expression levels of Imd-responsive AMPs, perhaps as a way to avoid transcribing genes that do not contribute to the animal's survival of the Gram-positive *E. faecalis* infections. However, the Imd-responsive AMPs were not down-regulated in a priming-specific manner (S2C and S2D Fig).

To further explore the role Imd signaling plays in a primed immune response, we tested survival of an *imd* mutant [16] to single and double injections (Figs 3F and 3G and S2E and S2F). As has been previously shown, the *imd* mutant showed a dose dependent response to *E. faecalis* infection with levels of lethality similar to a non-immunocompromised OreR control (Fig 3F & S1 Table; OreR HRs [Low Dose = 1.4 (0.97–2.1), High Dose = 5.7 (3.9–8.3)] & *imd* HRs [Low Dose = 1.3 (1.0–1.8), High Dose = 3.3 (2.4–4.6)]). However, when subjecting the flies to dual injections, we observed a significant, though not total, loss of priming ability in these *imd*-mutant flies (Fig 3G). *Efae*-primed flies still survive a second injection more effectively than Mock-primed flies (*Efae*-Primed vs. Mock-Primed HR = 0.39 [0.27–0.58], p = 2.5e-6), but less successfully than control flies twice injected with sterile PBS (*Efae*-Primed vs. PBS/PBS HR = 4.1 [2.7–6.3], p = 5.3e-12). This is in contrast to wildtype OregonR flies, which survive a secondary Efae infection as well as a double wounding (OreR Efae-Primed vs. PBS/PBS HR = 0.75.) We further probed the role of the Imd pathway in immune priming and found mutants in three additional pathway components, *kenny*, *Tab2*, and *Relish*, also show diminished immune priming, as indicated by HR > 1 when comparing the Efae-primed flies to the PBS/PBS control (S4 Fig). Together, this demonstrates that while the loss of the *imd* does not impact the survival of the flies with a single bacterial infection, it does negatively impact survival in animals that have been infected more than once.

## The hemocytes of primed animals up-regulate metabolic and translational pathways

Using the same approach as in fat bodies, we determined priming-specific transcription in adult hemocytes (S5A fig, full list of up-regulated and down-regulated genes in S5 Table). Hemocytes have several roles in the immune response, including bacterial phagocytosis, pathogen sensing, and signaling. Compared to fat bodies (Fig 3B), hemocytes showed a low amount of priming-specific up-regulation, with only 17 genes specifically up-regulated in the *Efae*-primed condition (Figs 4A and S5B). Most of these genes are poorly characterized or functionally unrelated (S5 Table). There were also 458 genes specifically up-regulated in animals with a single *Efae* High dose infection, indicating that the hemocyte transcriptional response to *E. faecalis* infection depends on the dose, previous injection state, and age of the animal. A GO term analysis reveals that many of these high dose specific genes are involved in immune response, as expected, and regulation of metabolic processes (S5C Fig). This analysis indicates that, in contrast to the fat body, hemocytes only upregulate a small number of genes specifically in the primed condition.

Similar to the fat body analysis, we identified hemocyte "core" genes as the up-regulated genes in all four conditions–animals with a single low dose infection, a single high dose infection, a mock-priming protocol, or a *Efae*-priming protocol. Of the 17 hemocyte core genes, 11 of them (~64%) overlapped with the 40 core genes found in fat bodies (S5D Fig and S5 Table). Among these were several Bomanins, *Drosomycin*, *SPH93*, *IBIN*, and *Metchnikowin-like*, implying a role for these genes in response to *E. faecalis* infection in both hemocytes and fat body. As with our fat body data, we again separately analyzed the levels of expression of the AMPs versus Bomanin effectors for hemocytes. When comparing expression levels of the core

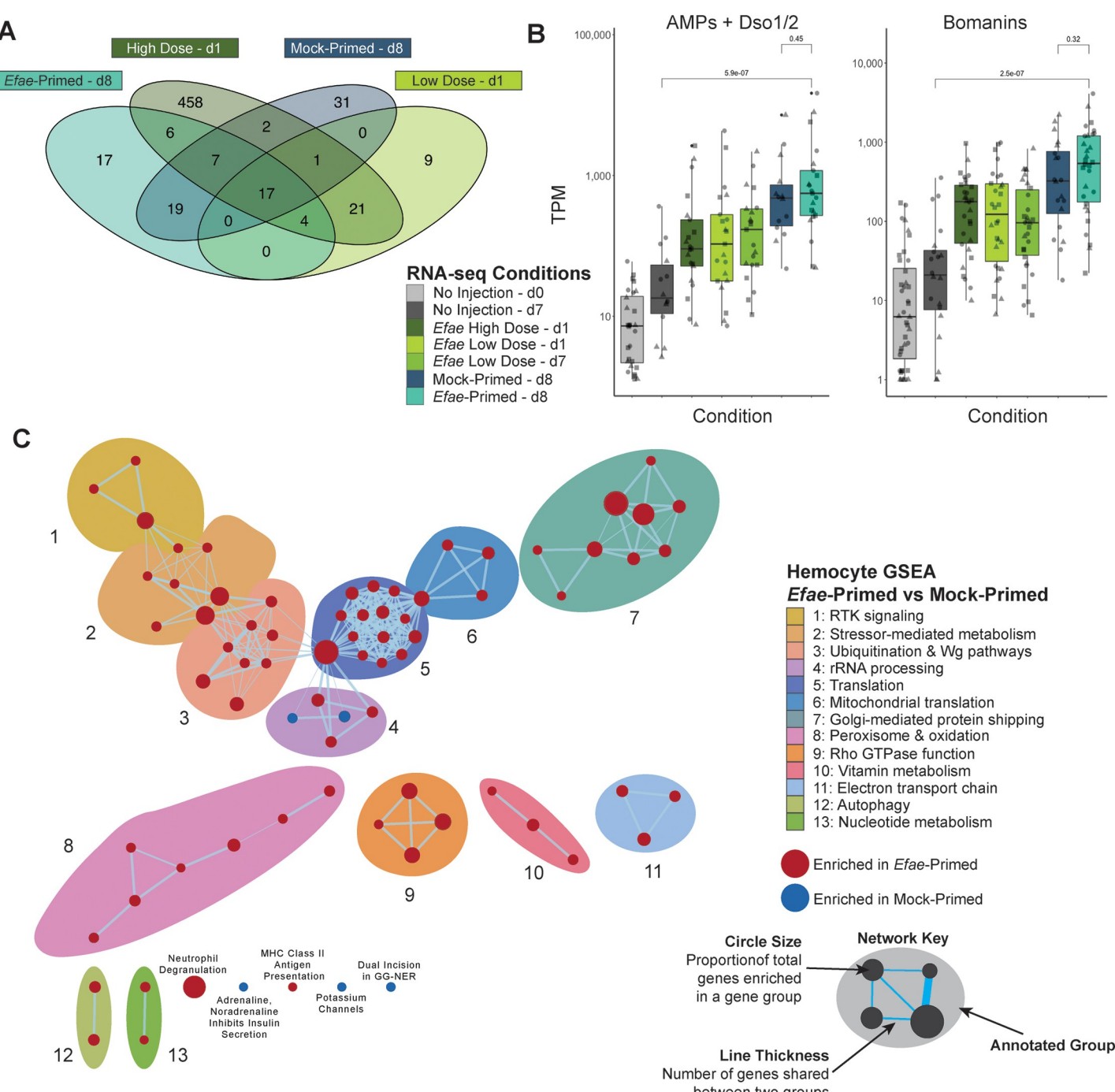

**Fig 4. Hemocytes do not significantly increase effector expression when primed, but differentially activate metabolic pathways.** A). Venn diagram of significantly up-regulated (log₂FC >1 & FDR <0.05) genes for hemocytes collected in the same conditions as Fig 3A). B). Expression in log₁₀ (TPM+1) of ubiquitously up-regulated AMPs [left] and *Bomanins* [right]. Condition colors match conditions in Fig 3A. Biological replicates are designated by the shape of individual points. While there is a significant difference in AMP and *Bom* expression in *Efae*-Primed hemocytes compared to their age-matched, non-injected controls (Wilcoxon test; AMPs: p = 5.9E-07, *Boms*: p = 2.5E-07), there is not a significant difference in ubiquitous AMP and *Bom* expression between Mock-Primed and *Efae*-Primed fat bodies (Wilcoxon test; AMPs: p = 0.45, *Boms*: p = 0.32). P-values were corrected for multiple testing using the Bonferroni correction. C). Network representation of Gene Set Enrichment Analysis (GSEA) for *Efae*-Primed versus Mock-Primed hemocytes. This visualization represents relationships between statistically significant terms (FDR < 0.05), manually curated with clusters that summarize the relationships between terms. Each circle represents an enriched gene set, circle size represents the relative proportion of genes within a set that were enriched, and the line thickness represents the number of genes that are enriched between any two gene sets. Full results are found in S6 Table.

*Boms*, we found no significant difference in expression between the Mock-primed and *Efae*-primed flies (Wilcoxon test: p = 0.32) (Fig 4B, right). Likewise, a comparison of the expression levels for the core AMP genes yielded no significant difference between the Mock-primed and *Efae*-primed flies (Wilcoxon test: p = 0.45) (Fig 4B, left). This indicates that, similar to the comparison between *Efae*-primed and Mock-primed fat bodies, transcripts associated with bacterial resistance are not specifically up-regulated in primed hemocytes.

Given the diverse functions of hemocytes in immune response, we decided to use GSEA to again systematically delineate priming-enriched pathways (Fig 4C, full GSEA analysis in S6 Table). Fig 4C shows individual gene sets enriched in either *Efae*-primed or Mock-primed hemocytes as nodes whose size represents the proportion of genes within a set that were found to be enriched. Edges (lines) connect nodes that share overlapping genes between gene sets, and their thickness represents how many genes are shared. This analysis of hemocyte transcription in *Efae*-primed samples versus Mock-primed samples indicated a wider picture of metabolic reprogramming (Clusters 2, 6, 8, 10, 11, and 13) and altered protein production (Clusters 4, 5, 6, and 7) in the primed samples. There was also enrichment for genes involved in antigen-presenting and neutrophil degranulation functions in mammalian orthologs, which contained several lysosomal and metabolic genes associated with bacterial immune response, such as the GILT family of genes.

## Several Toll effectors continuously express into re-infection

We further leveraged our transcriptomic data to identify genes that continuously express from the first infection into reinfection (Fig 5A). We defined continuously expressing genes as those that were up-regulated both 1 day and 6 days after a low dose infection (*Efae* Low-d1 & *Efae* Low-d7) and 1 day after the subsequent high dose infection (*Efae*-Primed-d8). Fat bodies had 14 genes that were identified as continuously expressing (Fig 5B), while hemocytes only had two (Fig 5C). For fat bodies, 13 of the 14 (~93%) continuously expressing genes overlapped with the identified core *E. faecalis* response genes (Fig 3B and 3C; annotated in S3 Table). Most of these genes are either known or suspected AMPs, and the list also includes a recently characterized lncRNA (lncRNA:CR33942) that can enhance the Toll immune response [34]. The fat body continuously expressing genes are largely Toll-regulated.

To further investigate the role Toll signaling is playing in creating a primed response to *E. faecalis*, we assayed infection response in flies with a *Myd88* mutation that eliminates intracellular Toll signaling and a *spz* mutant that eliminates extracellular Toll signaling (S6 Fig) [35]. In the single injection conditions, both mutants show the expected increased lethality when compared to our immune-competent control (S6A, S6C and S6E Fig) [4,31]. When assaying for survival against double-injected conditions, we found that *Myd88* mutants were still able to fully prime against *E. faecalis* re-infection with equivalent survival between the Efae-primed flies and the control flies injected twice with PBS (S6B Fig). However, the *spz* mutants lacked the ability to prime against *E. faecalis* (S6D Fig). The discrepancy between these two mutants requires further investigation (see Discussion).

## Potentiated gene expression plays a minor role in *E. faecalis* immune priming

In addition to priming-specific and continuously expressing genes, we also identified "recall response genes" [36]. These genes were defined as genes that are up-regulated in response to an initial low dose infection, turned off 6 days later, and up-regulated more strongly in response to a subsequent infection (Fig 5D). In fat bodies, we identified 7 recall genes (Fig 5E), and we did not identify any recall genes in hemocytes. Of these few fat body recall genes, we

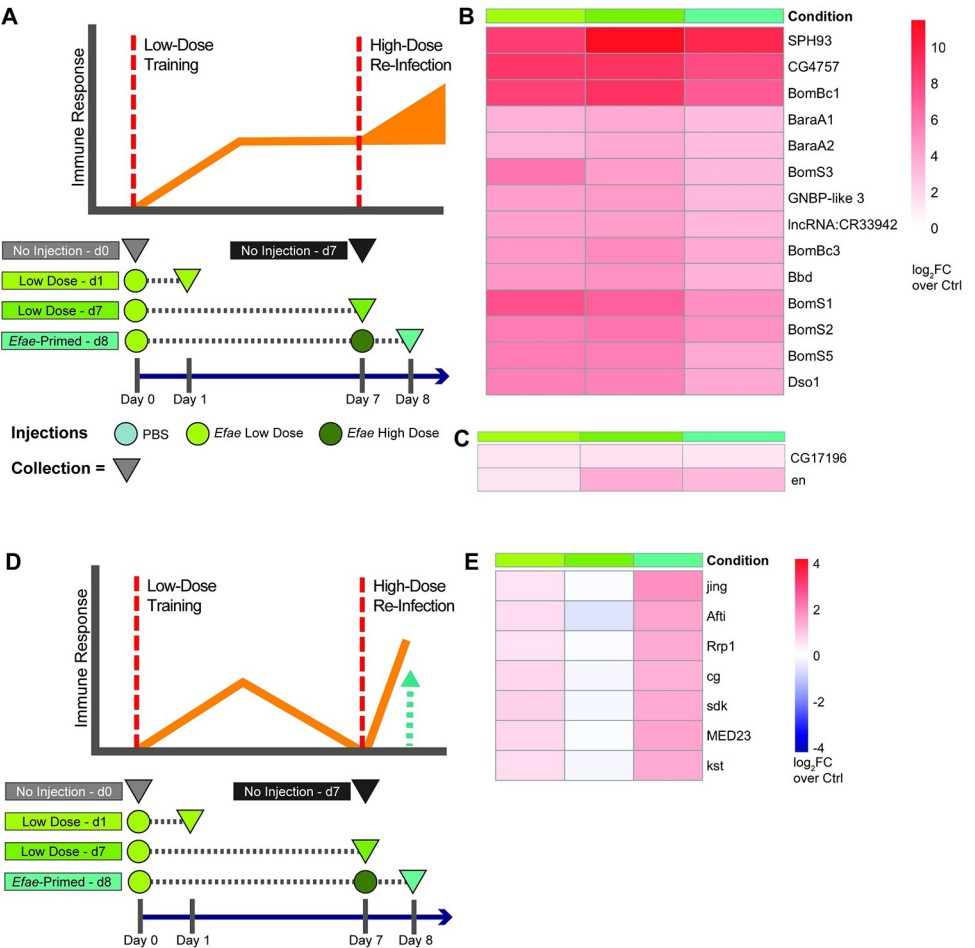

**Fig 5. Toll effector genes continuously express throughout *E. faecalis* immune priming, whereas few potentiated genes are recalled in *E. faecalis* immune priming.** A). Schematic of continuous gene expression from priming into re-infection. Experimental conditions are the same as Fig 1A, with the addition of age-matched, non-injected controls at Day 0 and Day 7 as well as an additional time point at Day 7 for collection of samples late in priming. Circles represent injections and triangles represent time of collection B). Continuously expressing genes in fat bodies (scale: log2FC over age-matched controls). C). Continuously expressing genes in adult hemocytes (scale: log2FC over age-matched controls). D). Schematic of immune recall response. E). Potentiated recall genes in fat bodies (scale: log2FC over age-matched controls).

found two Polycomb interacting elements (*jing* & *cg*) and a component of the Mediator complex (*MED23*), suggesting a potential role for transcriptional regulation. However, we did not find a strong role for recall transcription in our experiments.

## Discussion

In this study, we have shown the transcriptional underpinnings of a primed immune response against *Enterococcus faecalis* infection in *Drosophila melanogaster*. We demonstrated that a low dose of *E. faecalis* can prime the flies to better survive a high dose infection at least 7 days later, and the increase in survival is not linked to more effective clearance of the bacteria, but to increased tolerance of *E. faecalis* in primed animals. When comparing *Efae*-primed and Mock-primed animals, we found that the transcriptional profiles of antimicrobial peptides and *Bomanins* do not differ between the two conditions in either the fat body nor the hemocytes,

indicating that their differential expression is not driving survival in primed animals. However, there are ample transcriptional differences between the conditions, and GSEA analysis points to differences in cell cycle regulation and metabolic response. When testing priming ability in *imd* mutants we found that these mutants have unexpected survival phenotypes in the double injection conditions–*imd* mutants prime less effectively than wild type flies despite the dispensability of the pathway in response to a single infection.

Overall, we have seen evidence for tolerance, phagocytosis, and transcriptional reprogramming as drivers of priming against *E. faecalis* infection. Flies primed against *E. faecalis* re-infection did not actively clear bacteria more efficiently than Mock-primed controls (Fig 2B) and did harbor a higher bacterial load upon death (Fig 2C), both hallmarks of infection tolerance. We also found that phagocytosis was needed to fully prime, as supported by the decrease in priming ability in both bead-blocking experiments (Fig 1E) and *eater*-deficient flies (Fig 1F). Given that primed flies seem to survive infection by tolerating, rather than clearing bacteria, this suggests a role for phagocytes in priming other than their canonical responsibility of eliminating pathogens. One possibility is that phagocytes are working to sense an infection and relay that signal to other tissues through functional reprogramming [37,38]. This is supported by the large transcriptional shift in metabolic pathways seen in hemocytes, and specifically, the up-regulation of lysozyme-related pathways, including the "MHC Class II Antigen Presentation"and "Neutrophil Degranulation" gene sets (Fig 4C). Explicit proof of phagocyte reprogramming as a potential mechanism of priming merits further investigation.

Transcriptionally, there are three primary mechanisms suggested that may underlie immune priming–(1) primed animals may drive a qualitatively different expression program than mock primed flies, differentially regulated distinct genes, (2) primed flies may continually express key immune genes between a priming and subsequent infection, or (3) primed flies may re-active an immune response more quickly than unprimed flies. Our transcriptional data shows that most priming differences in both fat bodies and hemocytes can be attributed to gene expression that is unique to priming (Figs 3B & 4A). We saw continuous expression of a small number of Toll effectors in fat bodies (Fig 5B), and very little evidence of potentiated gene expression (Fig 5E).

There are previous studies of immune priming in flies, which taken together with this work paint a more complete picture of the phenomenon. One of the early descriptions of immune priming in *D. melanogaster* found a phagocytosis-dependent, AMP-independent priming response against *Streptococcus pneumoniae* [16]. Our study uses a different Gram-positive microbe, but a similar re-infection timescale. Similar to that study, we find that phagocytosis is needed to mount a primed immune response, as was demonstrated by the impaired priming in bead-blocked flies and *eater* mutants. (Separately, our *eater* mutants also showed increased survival in response to double-wounding alone, which indicates that either the mutation or the genetic background confers increased baseline survival to repeated wounding.) We also corroborated that survival is not correlated with AMP production. However, Pham et al. found that primed flies resist *S. pneumoniae* more effectively than naive flies, while our *Efae*-primed flies appeared to rely on immune tolerance to enhance survival. It is possible that this difference is due to the increased virulence of the pathogen, *S. pneumoniae*, which can kill a wild type fly with a relatively low dose of 3,000 CFU, compared to *E. faecalis*. The difference could also be due to the specificity of the host's primed response to different pathogens. More recent work also studied priming mechanisms in flies infected with *M. luteus* and *S. typhimurium* and found evidence of resistance and tolerance mechanisms, respectively [39]. *M. luteus* primed flies show potentiated gene expression upon re-infection. Prakash and co-workers have probed priming using *P. rettgeri* and found that a host of factors, including sex and

infection route can also shape immune priming [40]. In sum, these findings suggest that there may be multiple, bacteria-specific priming mechanisms.

Another study found that sterile wounding 2 days, but not 7 days, prior to infection with *E. faecalis* conferred some level of ROS-mediated protection [18]. This study's assay most closely matches our Mock-primed re-infections, and we also did not see enhanced survival when the wounding occurred 7 days prior to the infection. This indicates that the protection conferred from sterile wounding is effective in the short-term (i.e., 2 days), but not in the long-term (i.e. 7 days). However, both this study and our observations support the idea that hemocytes activate new functions in response to prior stimuli exposure (as was found in [16], as well). Finally, a study looking at the effects of chronic bacterial infection did not find immune priming with *E. faecalis* when using the same re-injection time points [12]. However, in that study flies were injected with two low-doses (~3,000 CFU/fly) and injected first in the abdomen and second in the thorax. This suggests a dose-dependent and/or injection site-dependent effect on priming ability.

One of the most surprising findings of this study is the priming responses found in the *imd*, *Myd88*, and *spz* mutant flies. As others have previously reported, our work demonstrates that the disruption of *imd* does not affect the fly's survival against a single low dose infection of *E. faecalis*. This is consistent with the well-described sensing of Gram-positive bacteria via Toll signaling and Gram-negative bacteria via Imd signaling [1]. However, we find that *imd* mutants lose some, though not all, of their priming capacity. The requirement of *imd* for survival was surprising for two reasons: first because Imd signaling has not been implicated in the survival of Gram-positive bacteria (or priming, in the case of *S. pneumoniae* in [16]), and second, because we saw down regulation of the *imd* gene in the fat body primed transcriptome. This suggests while downregulation of *imd* may be useful in priming, complete eradication of the pathway in the animal removes some priming ability. This could be due to the role the Imd pathway plays in modulating other key immune response pathways such as JAK/STAT, JNK, and MAPK signaling [2].

We were also surprised to see the variable role of Toll signaling for priming. Toll signaling plays a key role in surviving Gram-positive infections, and virtually all the persistently expressed genes we found here are known Toll targets (Fig 5B). While both Toll pathway mutants, *Myd88* and *spz*, showed markedly worse survival in response to a single low *E. faecalis* dose, they showed opposite effects in their ability to prime. Further work is needed to discern whether these genes' distinct molecular roles or the differences in genetic background between mutants account for the differences in priming ability.

While our data did not indicate a difference in bacterial clearance between *Efae*-primed and Mock-primed flies (Fig 2B), we acknowledge the possibility that the number of bacteria remaining in the animal from the initial infection may affect priming responses. As has been previously noted [11], we found variability in the bacterial burden during the initial low dose infection, consistent with some flies more effectively resisting infection than others (Fig 2A). Chronic infections tend to lead to low-level activation of the immune response throughout the animal's lifetime, causing expression of immune effectors that can loiter into re-infection and may contribute to enhanced survival [12]. It is not yet clear what effect the intensity of a chronic infection would have on priming ability, but it should be considered in the future. It is possible that a more severe chronic infection could either put the animal in a heightened state of "readiness" for a new infection or exhaust its resources.

Our data implies a major role for metabolic reprogramming in mediating a primed immune response against *E. faecalis*. Given the high energetic cost of mounting an immune response, it is logical to imagine immune priming as a more efficient re-allocation of metabolic resources to fine tune an immune defense strategy in a short-lived animal (as discussed in

[14,41]). Interestingly, evidence of metabolic shifts was not just relegated to the fat body (S3 Fig), which acts as the site of integration for metabolic and hormonal control, but was found to be the case with hemocytes, as well (Fig 4C). Similarly, in mammalian trained immunity where metabolic reprogramming drives epigenetic changes in innate immune cell chromatin [42]. Further characterization of *Drosophila* immune priming could explore the extent of differential metabolite usage when mounting a primed immune response and whether the transcriptional differences observed are encoded through epigenetic reprogramming of histone mark deposition, akin to what is observed in mammalian systems. Our study lays the groundwork for understanding the interplay between a physiological primed immune response and the transcriptional regulatory logic defining it.

## Methods

### Ethics statement

Studies were approved by the Institutional Biosafety Committees of the University of California Irvine and Boston University (protocol numbers 2015–1511 and 21–2522, respectively).

### Fly strains and Husbandry

Experiments, unless otherwise indicated, were performed using 4-day old Oregon-R male flies. *Eater* mutants are described in [22] and were obtained from the Bloomington Stock Center (RRID:BDSC_68388). These flies knocked out the *eater* gene through homologous recombination that replaced 745bp of the TSS, exons 1 and 2, and part of exon 3 with a 7.9 kb cassette carrying a $w^{[+]}$ gene. The $imd^{1091}$ line, the *w; key*[1]*, cn, bw; gIKKγ*[WT] line, *Tab2*[AOII3] line, and the *Rel*[E20] line were provided by Neal Silverman. The $imd^{1091}$ mutants were generated by creating a 26bp deletion at amino acid 179 that creates a frameshift mutation at the beginning of the death domain in *imd* [16]. *Myd88*[kra-1] flies were provided by Steve Wasserman and Lianne Cohen. This line was created by excising 2257bp of the *Myd88* gene spanning the majority of the first exon and inserting a P-element [35]. Stable lines were balanced against a CyO balancer with homozygous mutant males being selected for injections. *Spätzle* mutants were obtained from the Bloomington Stock Center ($spz^2ca^1$/TM1, RRID:BDSC_3115). Stable lines were balanced against a TM1 balancer with homozygous mutant males being selected for injections. Flies were housed at 25˚C with standard humidity and 12 hr-light/12 hr-dark light cycling.

### Injections

All bacterial infections were done using a strain of *Enterococcus faecalis* originally isolated from wild-caught *Drosophila melanogaster* [19]. Single colony inoculums of *E. faecalis* were grown overnight in 2mL BHI shaking at 37˚C. 100uL of overnight *E. faecalis* inoculum was then added to 2mL fresh BHI and grown shaking at 37˚C for 2.5 hours before injections to ensure it would be in the log-phase of growth. Bacteria was then pelleted at 5,000 rcf for 5 minutes, washed with PBS, re-suspended in 200uL PBS, and measured for its OD600 on a Nanodrop. Flies were injected with either PBS, *E. faecalis* at OD 0.05 for low dose experiments (~3,000 CFU/fly), or *E. faecalis* at OD 0.5 for high dose experiments (~30,000 CFU/fly). Due to the high heat resistance of *E. faecalis*, heat-killed inoculums were produced by autoclaving 10mL cultures that were in log-phase growth. Successful heat-killing was determined by streaking 50uL on a BHI plate and checking it had no growth. Adult flies were injected abdominally using one of two high-speed pneumatic microinjectors (Tritech Research Cat. # MINJ-FLY or Narishige IM 300) with a droplet volume of ~50nL for both PBS and bacterial

injections. Injections into a drop of oil on a Lovin's field finder were used to calibrate the droplet volume. Injections were performed in the early afternoons to control for circadian effects on immune response. Flies were not left on the $CO_2$ pad for more than 10 minutes at a time. Injected flies were housed in vials containing a maximum of 23 flies at 25˚C with standard humidity and 12 hr-light/12 hr-dark light cycling.

## Survival assays

To track survival, flies were observed every 24 hours at the time they were injected. Media was changed every three days with flies being exposed to $CO_2$ for no more than two minutes between vial transfers. Survival is plotted as Kaplan-Meier curves using the R 'survival' and 'survminer' packages. Cox proportional hazards were used to compare survival experiments. Comparisons on survival between two conditions is presented as a hazard ratio (HR) that scores survival rate of a test group against survival in a referent group. A HR is reported with its 95% confidence interval and a Wald test p-value with Benjamini-Hochberg correction for multiple comparisons reporting whether the HR significantly deviates from 1.

## Bead blocked infection

To ablate phagocytosis during the initial low dose infection, flies were first abdominally injected with 50nL Cml latex beads (Thermo Scientific Cat. # C37480), allowed to rest for 4 hours, and then injected with ~3,000 *E. faecalis*. Primed survival was then assayed for after injection with ~30,000 CFU of *E. faecalis* 7 days after the initial bacterial infection (as was previously described in Pham, et al. 2007).

## Dilution plating

Single flies were suspended in 250uL PBS and homogenized using an electric pestle. The homogenate was then serially diluted five-fold and plated on BHI plates and left to grow in aerobic conditions for two days at 25˚C. Using this method there was little to no background growth of the natural fly microbiome. Images were then taken of each plate using an iPhone XR and analyzed using ImageJ with custom Python scripts to calculate colony forming units (CFU) per fly. Plotting was done using the R package ggplot2 [43]. Comparisons between bacterial loads were done using rank-sum tests on log-transformed data.

## Hemocyte isolation

For each biological replicate, 20 flies were placed in a Zymo-Spin P1 column with the filter and silica removed along with a tube's-worth of Zymo ZR Bashing Beads. Samples were centrifuged at 10,000 rcf at 4˚C for one minute directly into a 1.5mL microcentrifuge tube containing 350uL TriZol (Life Technologies) (schematic in S5A Fig). Samples were then snap frozen and stored at -80˚C for future RNA extraction.

## Fat body isolation

Each biological replicate consisted of 3 extracted fat bodies. Flies were anesthetized with $CO_2$ and pinned with a dissection needle at the thorax, ventral side up, to a dissection pad. The head, wings, and legs were then removed using forceps. Using a dissection needle, the abdomen was carefully opened longitudinally, and the viscera removed using forceps. The remaining abdominal filet with attached fat body cells was then removed from the thorax and transferred to a 1.5mL microcentrifuge tube on ice containing 350uL TriZol. Samples were then snap frozen and stored at -80˚C for future RNA extraction. Dissection of fat bodies

includes some level of testes and sperm contamination, which was monitored by tracking expression of sperm-related genes in RNA-seq libraries and throwing out any libraries that have relatively high expression of said genes (S7 Fig).

### RNA-seq library preparation

RNA from either fat bodies or hemocytes was extracted using a Zymo Direct-zol RNA Extraction kit and eluted in 20uL water. Libraries were prepared using a modified version of the Illumina Smart-seq 2 protocol as previously described [26]. Libraries were sequenced on an Illumina Next-seq platform using a NextSeq 500/550 504 High Output v2.5 kit to obtain 43bp paired-end libraries.

### Differential gene expression analysis

Sequenced libraries were quality checked using FastQC and aligned to *Drosophila* reference genome dm6 using Bowtie 2 [44]. Counts were generated using the subread function feature-Counts. Counts were then loaded into EdgeR [45], libraries were TMM normalized, and genes with CPM < 1 were filtered out. Full code used in downstream analysis can be found at https://github.com/WunderlichLab/ImmunePriming-RNAseq.

### Priming-specific transcription analysis

To identify priming-specific up-regulation, we first identified genes that were significantly up-regulated ($\log_2$FC>1 & FDR<0.05) in each condition that assayed for immune response 24 hours after infection (i.e. *Efae* Hi Dose-d1, *Efae* Low Dose-d1, Mock-Primed-d8, and *Efae*-Primed-d8) (the effect of modulating significance and $\log_2$FC cut-offs can be seen in S8 Fig). These gene lists were then compared to each other for overlap. Genes that were only up-regulated in *Efae*-Primed-d8 samples, but in no other condition were labeled as "priming-specific". Average expression of AMPs and *Bomanins* plotted as a box-and-whisker plot of $\log_{10}$(TPM +1) to show variance. Significant differences between conditions were calculated using a Wilcoxon rank sum test using a Bonferroni correction for multiple comparisons.

### Continuous expression analysis

To determine genes that were continuously being expressed throughout initial immune priming into re-infection, we focused on the transcription in samples assayed at *Efae* Low-d1, *Efae* Low-d7, and *Efae*-Primed-d8. We first selected genes that were expressed at the above time points relative to a non-stimulated, age-matched control ($\log_2$FC >0). We then filtered that shortlist on the following conditions: genes had to significantly up-regulated at *Efae* Low-d1 compared to its age-matched control ($\log_2$FC>0 & FDR<0.05), genes had to significantly up-regulate at *Efae*-Primed-d8 compared to its age-matched control ($\log_2$FC>0 & FDR<0.05), and genes had to either stay at similarly expressed levels or increase in expression between *Efae* Low-d7 and *Efae*-Primed-d8 compared to their age-matched controls ($\log_2$FC≥0).

### Potentiated recall response analysis

We termed genes as being "recalled" if they were initially transcribed during priming (*Efae* Lo-d1 $\log_2$FC over age-matched control > 0.5), ceased being expressed by the end of priming (*Efae* Lo-d7 $\log_2$FC over age-matched control ≤ 0), and were then re-expressed upon re-infection (*Efae*-Primed-d8 $\log_2$FC over age-matched control > 0.5 & FDR < 0.1). Our significance threshold had to be somewhat relaxed for expression after re-infection to detect any recalled gene expression at all. To delineate genes that were truly re-activating transcription in a

potentiated manner (i.e., at a higher level upon re-infection as compared to when they were initially expressed during priming), we also filtered on the conditional that $\log_2$FC over age-matched controls had to be higher in *Efae*-Primed-d8 versus *Efae* Low-d1. Finally, to identify genes that were recalled only in our primed samples, we further filtered on the condition that genes had to have a $\log_2$FC $\leq 0$ over age-matched controls for Mock-primed-d8 samples.

## GO term enrichment

All GO Term Enrichment was done using Metascape's online tool [46] and plotted using custom ggplot2 scripts.

## Gene set enrichment analysis

Gene set enrichment analysis was run using the GSEA software v. 4.2.3 [27]. *Drosophila*-specific gene matrices for both KEGG and Reactome-based GSEA aliases were taken from [47]. TMM-normalized TPMs were extracted from EdgeR analysis and used as input for two-condition comparisons using GSEA software. Due to the low number of replicates ($< 7$ replicates per condition), analysis was run using a gene set permutation. Full tabular results are found in S4 and S6 Tables. An enrichment map visualizing the network of enriched gene sets was created using Cytoscape (Node Cutoff = 0.1 FDR; Edge Cutoff = 0.5) and clusters describing the mapping manually curated [48].

## Supporting information

**S1 Fig. Dynamics of *E. faecalis* priming and double-injection survival in OreR flies.** A). Survival is similar in flies allowed to prime with a low-dose of *E. faecalis* (~3,000 CFU/fly) for varying amounts of time before re-infection with a high dose of *E. faecalis* (~30,000 CFU/fly) (n: 1 Day = 38, 2 Days = 22, 4 Days = 54, 6 Days = 63, 7 Days = 78). B). There is a significant difference in survival (log-rank sum test, p<0.0001) in OreR flies injected once with PBS (PBS, n = 149) or twice with PBS with seven days of rest between repeated injections (PBS/PBS, n = 74). Dotted lines indicate median survival time; shaded regions indicate 95% confidence intervals. C). There is a significant difference in survival (log-rank sum test, p = 0.0063) in OreR flies injected once with a high dose of *E. faecalis* (*Efae* High, ~30,000 CFU/fly, n = 74) versus primed with a low dose of *E. faecalis* for seven days and then re-infected with a high dose of *E. faecalis* (*Efae*-Primed, n = 78). Data are the same as Fig 1B and 1C, replotted for comparison.
(TIF)

**S2 Fig. Additional analysis for fat body RNA-seq.** A). Heatmap of $\log_2$FC over non-injected controls of whole-body, core immune response genes from Troha, et al. 2018. Only a subset of the core genes were ubiquitously expressed across all conditions assayed for in this study. The differences are likey due to distinctions in time point and tissue. B). GO term enrichment from fat body *Efae*-Primed-specific [top] and Mock-Primed-specific [bottom], up-regulated genes. C). Expression in $\log_{10}$(TPM+1) of IMD-dominant AMPs. Biological replicates are designated by the shape of individual points. While there is a significant difference in IMD AMP expression in *Efae*-Primed fat bodies compared to their age-matched, non-injected controls (Wilcoxon test; p = 6.3E-05), there is not a significant difference in expression between Mock-Primed and *Efae*-Primed fat bodies (Wilcoxon test; p = 0.37). D). Heatmap of $\log_2$FC over non-injected controls of IMD-dominant AMPs across collected fat body samples E). Single-injection survival comparison between OreR and *imd*-mutant flies F). Double-injection survival comparison between OreR and *imd*-mutant flies. Data from D & E are the same as in

 replotted for comparison.
(TIF)

**S3 Fig. GSEA for *Efae*-primed versus Mock-primed fat bodies.** This visualization represents relationships between statistically significant terms (FDR < 0.05), manually curated with clusters that summarize the relationships between terms. Full results are found in S4 Table.
(TIF)

**S4 Fig. Single- and double-injection survival for additional IMD pathway mutants.** A). Survival of single-injected Tab2 mutant flies versus PBS control (PBS: n = 143, Efae Low Dose: n = 114, Efae High Dose: n = 67). Dotted line indicates median survival time. Shaded area indicates 95% confidence interval. Low Dose vs PBS: HR = 1.1, p = 0.76; High Dose vs PBS: HR = 3.4, p = 5.6E-08; pairwise comparisons are calculated using a Cox proportional hazard model with hazard ratios and Wald statistic values reported for experimental conditions versus their PBS negative control; significance values are adjusted for multiple testing using a Benjamini-Hochberg method. B). Survival of primed Tab2 mutant flies versus double-injected, non-primed controls (PBS/PBS: n = 63, Mock-Primed: n = 70, Efae-Primed: n = 65). Efae-Primed vs PBS/PBS: HR = 2.0, p = 0.0016; Mock-Primed vs. PBS/PBS: HR = 4.6, p = 1.1E-10. C). Survival of key mutant flies injected with PBS (n = 155), Efae Low Dose (~3,000 CFU/fly, n = 148), and Efae High Dose (~30,000 CFU/fly, n = 69). Low Dose vs PBS: HR = 2.2, p = 7.5E-05; High Dose vs PBS: HR = 5.3, p = 2.0E-16 D). Survival of primed key mutant flies versus double-injected, non-primed controls (PBS/PBS: n = 75, Mock-Primed: n = 60, Efae-Primed: n = 71). Efae-Primed vs PBS/PBS: HR = 2.3, p = 3.6E-05; Mock-Primed vs. PBS/PBS: HR = 7.1, p = 3.5E-14. E). Survival of single-injected Rel mutant flies versus PBS control (PBS: n = 140, Efae Low Dose: n = 63, Efae High Dose: n = 60). Low Dose vs PBS: HR = 0.57, p = 0.0014; High Dose vs PBS: HR = 2.8, p = 7.2E-08. F). Survival of primed Rel mutant flies versus double-injected, non-primed controls (PBS/PBS: n = 55, Mock-Primed: n = 64, Efae-Primed: n = 56). Efae-Primed vs PBS/PBS: HR = 3.7, p = 3.1E-09; Mock-Primed vs. PBS/PBS: HR = 5.7, p = 1.2E-13. Like *imd* mutants, *Rel*, *key*, and *Tab2* mutants lost the ability to fully prime against *E. faecalis* infections. The relative severity of the loss does depend on the mutant, possibly due in part to differences in genetic background, with *Relish* mutants showing the weakest priming ability, followed by *key*, and then *Tab2* (which shows only a minor priming defect).
(TIF)

**S5 Fig. Additional data for hemocyte RNA-seq.** A). Schematic diagram of hemocyte RNA extraction. B). Significantly up-regulated genes as corresponding to conditions in Fig 4A (scale: $log_2$FC over age-matched controls). C). GO term enrichment from hemocyte *Efae* Hi Dose-specific, up-regulated genes. D). Overlap of up-regulated core genes (4-condition overlap in Venn diagram) between hemocytes and fat bodies. Created with Biorender.
(TIF)

**S6 Fig. Additional data for continuous expression RNA-seq.** A). Survival of single-injected *Myd88* mutant flies versus PBS control (PBS: n = 135, Efae Low Dose: n = 107, *Efae* High Dose: n = 67). Low Dose vs PBS: HR = 4.3 [2.8–6.6]; High Dose vs PBS: HR = 13 [8.8–22]. B). Survival of primed *Myd88* mutant flies versus double-injected, non-primed controls (PBS/PBS: n = 60, Mock-Primed: n = 69, *Efae*-Primed: n = 60). *Efae*-Primed vs PBS/PBS: HR = 0.56 [0.34–0.92]; Mock-Primed vs. PBS/PBS: HR = 1.8 [1.2–2.7]. C). Survival of single-injected *spz* mutant flies versus PBS control (PBS: n = 64, *Efae* Low Dose: n = 65, *Efae* High Dose: n = 74). Low Dose vs PBS: HR = 2.9 [1.9–4.5]; High Dose vs PBS: HR = 6.1 [4.1–9.1]. D). Survival of primed *spz* mutant flies versus double-injected, non-primed controls (PBS/PBS: n = 69, Mock-

Primed: n = 81, *Efae*-Primed: n = 50). *Efae*-Primed vs PBS/PBS: HR = 4.0 [2.6–6.1]; Mock-Primed vs. PBS/PBS: HR = 3.4 [2.3–5.1]. E). Single-injection survival comparison between OreR and *Myd88*-mutant flies. F). Double-injection survival comparison between OrR and *Myd88*-mutant flies. Data are the same as in A & B, replotted for comparison.
(TIF)

**S7 Fig. Quality control of fat body RNA-seq libraries.** A). Spearman correlation heatmap of fat body RNA-seq libraries. Values are $R^2$ spearman correlation values. B). Expression of sperm motility genes in fat body RNA-seq libraries. Values are $\log_2(TPM+1)$.
(TIF)

**S8 Fig. Modulation of significance and fold-change cutoffs in differential analysis.** A). Overlap analysis between 24-hour RNA-seq in fat bodies when changing fold-change cut-offs along the y-axis ($\log_2FC>0$, $\log_2FC>1$, $\log_2FC>2$) and significance cutoffs along the x-axis ($FDR<0.1$, $FDR<0.05$, $FDR<0.01$). B). Same analysis as A, with hemocytes.
(TIF)

**S1 Table. Summary statistics for all survival curves calculated using Kaplan-Meier visualizations and Cox proportional hazard modeling.**
(XLSX)

**S2 Table. Sequencing information for fat body and hemocyte RNA-seq.**
(XLSX)

**S3 Table. Lists of up-regulated genes specific to each fat body condition assayed in Fig 3, common between all fat body conditions, and specifically down-regulated in *Efae*-Primed-d8 fat bodies.**
(XLSX)

**S4 Table. Gene set enrichment analysis for *Efae*-Primed vs Mock-Primed fat bodies.** Clustering and terms are shown in S3 Fig. This represents the tabular output directly from the GSEA software v. 4.2.3 (Subramanian 2005).
(XLSX)

**S5 Table. Lists of up-regulated genes specific to each hemocyte condition assayed in Fig 4, common between all hemocyte conditions, specifically down-regulated in *Efae*-Primed-d8 fat bodies and overlap between common *Efae*-response genes in fat bodies and hemocytes.**
(XLSX)

**S6 Table. Gene set enrichment analysis for *Efae*-Primed vs Mock-Primed hemocytes.** Clustering and terms are shown in Fig 4C. This represents the tabular output directly from the GSEA software v. 4.2.3 (Subramanian 2005).
(XLSX)

## Acknowledgments

We would like to thank S. Wasserman, N. Silverman, and Bloomington Stock Center for fly strains; B. Lazzaro for bacterial strains; A. Mortazavi, C.J. McGill, and H.Y. Liang for access to their sequencing core and technical assistance with library preparation. We would like to thank L. Cohen and B. Ramirez-Corona for constructive discussion on this work.

## Author Contributions

**Conceptualization:** Kevin Cabrera, Zeba Wunderlich.

**Data curation:** Kevin Cabrera.

**Formal analysis:** Kevin Cabrera, Zeba Wunderlich.

**Funding acquisition:** Kevin Cabrera, Zeba Wunderlich.

**Investigation:** Kevin Cabrera, Duncan S. Hoard, Olivia Gibson, Daniel I. Martinez.

**Software:** Kevin Cabrera, Duncan S. Hoard.

**Supervision:** Zeba Wunderlich.

**Visualization:** Kevin Cabrera.

**Writing – original draft:** Kevin Cabrera, Zeba Wunderlich.

**Writing – review & editing:** Kevin Cabrera, Zeba Wunderlich.

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
