## [Decision Letter · Decision Letter 0]

31 Oct 2022

Dear Dr. Wunderlich,

Thank you very much for submitting your manuscript "Drosophila immune priming to Enterococcus faecalis relies on immune tolerance rather than resistance" for consideration at PLOS Pathogens. As with all papers reviewed by the journal, your manuscript was reviewed by members of the editorial board and by several independent reviewers. In light of the reviews (below this email), we would like to invite the resubmission of a significantly-revised version that takes into account the reviewers' comments.

We cannot make any decision about publication until we have seen the revised manuscript and your response to the reviewers' comments. Your revised manuscript is also likely to be sent to reviewers for further evaluation.

Sincerely,

Danielle A. Garsin

Guest Editor

PLOS Pathogens

Nina Salama

Section Editor

PLOS Pathogens

Kasturi Haldar

Editor-in-Chief

PLOS Pathogens

orcid.org/0000-0001-5065-158X

Michael Malim

Editor-in-Chief

PLOS Pathogens

orcid.org/0000-0002-7699-2064

Reviewer's Responses to Questions

**Part I - Summary**

Reviewer #1: In the manuscript entitled “Drosophila immune priming to enterococcus faecalis relies on immune tolerance rather than resistance” the authors develop a model for immune priming of Drosophila with Enterococcus faecalis and show that this results in tolerance of infection. They go on to conduct extensive RNAseq of the fat body and hemocytes at different time points in their infection models. They then subject IMD pathway, Toll pathway, and Eater mutants to their infection models and determine that IMD and Eater but not Toll play partial roles, suggesting that their priming phenotype is complex. Priming is not a new finding, and it is not clear to me that the minor differences the authors point out between these observations and those of previous investigators reflect differences in microbe pathogenicity as opposed to differences in numbers of bacteria injected and time to second challenge. While the extensive RNAseq data are likely to contain valuable information, they are not analyzed here with enough rigor or follow-up to yield testable hypotheses regarding the mechanism of priming. Furthermore, the language in the manuscript is not precise and clear, the figures are not referred to accurately, and the statistics need to be more carefully explained or corrected.

Reviewer #2: In this manuscript, Cabrera and colleagues use Drosophila as a model to study immune priming in insect. They find that flies primed with a low dose of E faecalis survive better a challenge with a high dose of E faecalis. They further try to analyze the mechanisms underlying such priming effect. They find that priming is not correlated with a change in bacterial growth in the fly, suggesting it is a tolerance problem rather than a resistance problem. They further show that priming requires phagocytosis and IMD pathway. This paper is very well written and easy to follow. Their transcriptomics data are well documented and overall the idea that imd could be involved in an increase in tolerance during priming intriguing. However, I do not feel that the few genes coming from transcriptomics that they single out at the end of the manuscript is the strongest conclusion to that story. I feel that this paper is interesting and has potential to be published in PLOS Pathogens. However, I would want that results involving phagocytosis and the requirement of IMD are strengthened, and that tolerance is measured in these conditions before supporting the message of this paper fully.

Reviewer #3: The manuscript present an analysis of the priming of the innate immune system of Drosophila melanogaster with a low dose – high doses E. faecalis model. The first evidence for this is a low dose of E. faecalis (a Gram-positive) improves the survival of adult animals against a second higher dose of bacteria. Further evidence indicates that this improvement is due to tolerance and not resistance, and priming requires the Imd pathway and phagocytic activity of hemocytes to work. On the other hand, The Toll pathway, itself responsible for the initial defense against E. faecalis, is not required for the priming. An RNAseq experiment reveals strong alteration in transcriptional profiles of fat body and hemocytes with priming and infection protocols, not surprisingly. The authors’ analysis of the RNAseq data claimed to reinforces the idea of tolerance being the driving mechanism of priming, but the underlying was not clear.. From the RNAseq data the authors identify genes that that stay upregulated throughout (“loiter”), which include several Toll pathway genes. The authors also identified some genes that turn off but get turned back on after the second infection (“recall”) but do not elaborate on their roles or function in priming.

**Part II – Major Issues: Key Experiments Required for Acceptance**

Reviewer #1: 1) Figure 1B and 1C: One of the consistent observations from these experiments is that mock-primed bacteria die much faster than bacteria that are simply given one injection of high-dose E. faecalis. This is supported by the data in Figure 2 showing that the bacterial burden of mock primed flies on day 1 is two orders of magnitude higher than that of flies administered high dose Efae without mock priming. This suggests that mock-priming makes the fly more susceptible to infection, while priming with low dose bacteria makes the fly less susceptible. Is this the case? If so, how does this inform the mechanism of priming?

2)

a) Here the authors graph expression of AMPs and Bomanins in the various conditions and comment that there is no difference between mock primed and Efae-primed. How was the synthesis of the data for all the time points and AMPs carried out? If the time points and AMPs were analyzed separately, would any significant differences be noted? Please give a rationale for lumping all the data together and describe how it was done.

b) Do these data come from the RNAseq data set? If so, it is not clear to me that a Welch’s test is the correct one to apply. The significance test applied should be justified.

c) The error in these measurements is quite large especially in Fig S2C. Is this a function of data aggregation? Does the unaggregated data yield a significant decrease in AMPs for Efae-primed flies at a particular time point, which would go along with the observed decrease in IMD in 1F? Even for the aggregated data, the trend is in that direction. Interestingly, based on 1F, one could also claim that there is no difference in IMD expression between uninfected and mock-primed flies, but the error bars are quite large. A follow-up qRT-PCR experiment might give more confidence in such statements.

Reviewer #2: MAJOR COMMENTS:

So far, I feel that the strength of the paper is to propose that phagocytosis and IMD pathways are involved in tolerance after priming. However, this currently is only supported by a mutant analysis in each case. Therefore I request some more data to support this model. Precisely, that means:

1- Get more than one mutant to demonstrate the requirement of phagocytosis. Additional hemocyte manipulation (genetic, ablation, using beads etc etc) is required.

2- Get more than one mutant to demonstrate the IMD pathways is indeed required. I would recommend one additional way to alter IMD protein, and maybe to test additional mutants/KDs for other members of IMD pathway proteins.

3- It is unclear that when phagocytosis or IMD are lacking, tolerance is affected. The only measure of CFUs is done in the wild type, and it s only one proxy at one timepoint which may still be resistance. I suggest that in addition to what is currently provided, the authors measure the bacterial load upon death as published in Duneau at al, 2017. Alternatively, they could actually perform a regression analysis with multiple doses of pathogens and survival (the method implemented by David Schneider).

4- They should redo their measure of tolerance in their mutants of phagocytosis and IMD pathway. This is unclear to me that these mutants actually lead to a change in tolerance, and this is inferred based on WT, but not demonstrated currently.

Reviewer #3: Figure 1C, 1E, 3H, 5E: The key comparison and conclusions in this study rely on qualitative examination of survival curves and median survival time (without statistical testing) to support the authors’ arguments. However, the actual statistical analysis, as presented, do not support these conclusions. The P-values from the log-rank test of survival curves indicate that all mutants have similar priming activity (usually with P-values marked with ***), which contradicts the main claims the authors make. The authors may be correct in their intuitive analysis of the data, but the statistics are inadequate for their arguments. For example, it does not appear that the log-rank analysis was corrected for multiple comparison which might result in an analysis consistent with the authors intuitive conclusions; the mean survival time should be statistically analyzed, and/or perhaps a quantitative analysis of the hazard ratio might be informative. Regardless, the statistical testing needs to support the main conclusions.

The wild type background of the survival experiments are not properly controlled. Additionally, one case (1C & 1E) it seems that the PBS/PBS groups have vastly different response. This undermines the ability of the authors to make claims about the contribution of each pathway and process toward priming. To be explicit, none of the mutant strains analyzed are in the Oregon R background; a matched wildtype needs to be included in all analyses.

The visual and labels unsed in figures 3 and 4 are clear on the conditions used in the RNAseq experiment. However, this text is completely muddled, when the authors attempt to describe the analysis and results, and this reviewer was unable to follow the narrative.

The authors also claim that hemocytes act as signal relayers for the priming action, but this claim is “synthesized” from two unrelated experiments and the authors do not attempt to explain how the phagocytosis activity of hemocytes can work as a signal relaying mechanism or how this might play into the improved tolerance which is suggested as key mechanism for priming. In general, the manuscript lacks a thorough mechanistic probing of the phenomenom presented.

The authors suggest that improved tolerance is responsible priming, but aside from bringing up metabolism in RNAseq analysis while discounting the possible roles of loitering or recall genes, the authors do not further attempt to explain or suggest a mechanism for bacteria tolerance

The GSEA visual and methodology is quite novel, and yet it is too complex and unclear. The authors should attempt to reduce the complexity and explain the meaning behind the analysis and the visualization, as well as how each claim is drawn.

**Part III – Minor Issues: Editorial and Data Presentation Modifications**

Reviewer #1: 1) Introduction, line 44: I am not sure what “qualitatively different” means here.

2) Figure 1 C and D: The Efae-primed survival data for Figs C and D look very similar. If they are from the same experiment, these graphs should be combined, or the authors should note in the Figure Caption that the same data were used in both figures.

3) Line 105, Fig 1E: The authors conclude that the differential response to live and dead bacteria suggests a mechanism other than bacterial sensing. Another possibility is that bacterial sensing is required throughout the seven-day priming interval, and the products of the dead bacteria are cleared rapidly and no longer activate bacterial sensors.

4) Line 109, Figure 1E: These data are referred to again at the end of the results section. The authors should consider moving all discussion of these data to the section at the end of results.

5) Line 140 and Fig 2B: A time course for bacterial burden is shown but only one p-value is given in the text. Which time point does this p value refer to? Also, it is most informative to indicate p values on the graph for all comparisons in all figures.

6) Line 167, Fig 1B: The fat body GO analysis terms highlighted refer to general processes such as immune response, response to stress, and cell surface receptor signaling. Without follow-up experiments, these categories are not particularly informative.

7) Figure 3: It is difficult to remember the color-coding scheme for the different experimental conditions studied. I suggest including a key in every figure in which the color coding is used.

8) Figure 3E and F and S2C:

9) Abstract, line 15 and throughout: The phrase “loitering genes” is vague. The genes are present in all cases. They are persistently differentially regulated. In line 49, the authors state the effectors loiter. Has the concentration of effectors been measured? I suggest the authors use terminology that describes the phenomenon in place of the term “loiter.”

10) Introduction, line 44: I am not sure what “qualitatively different” means here. Did the authors not measure the response?

11) Introduction, line 50: I would eliminate “often” here unless this has been shown to be a frequent state of flies.

12) Figures 2A and 1C are referred to out of order.

13) Line 271: Figure 3C should be Figure S3C. Furthermore, I do not see data for an Eater mutant in Figure 2B.

14) Line 280: Authors refer to Figure 1F, which does not exist. This should be Figure 1E.

15) Line 338: Myd88 mutants have no effect. This is not “an unexpected effect.” This should be rephrased.

Reviewer #2: MINOR COMMENTS

line 99: fig1C, Ef primed - PBS injected control is missing; what about other 'wt' strains? is this effect OrR specific or general? PBS/PBS effect seems to be dramatic for OrR (but not for Eater-/)

line 113: seems to be very similar to immune priming effect seen in OrR (fig1C)

OrR 50% lethality (Ef primed - Mock primed): 4d - 1d

Eater 50% lethality (Ef primed - Mock primed): 3d - 1d

OrR - Eater: is the difference significant?

line 116: “indicating that phagocytosis is needed to allow Efae-primed flies to survive”: i am not sure this is valid. Could the authors elaborate?

line 143: i think that BLUD data are needed to make this conclusion, or alternatively or more stringent measure of tolerance.

line 151: how different are these samples if we compare global gene expression data?

line 159: it would be useful to see gene names / clusters

line 228: i see difference between OrR and imd mutant: why do they conclude that imd makes no impact?

line 230: would it make sense to compare the survival after high dose Ef in G and high dose survival in H?

line 233: based on high dose Ef injection on 3G: it does. What do the authors think?

line 235: “This suggests that there are distinct differences in use of signaling pathways between animals with one versus two infections.” i am not sure about this conclusion. Could the authors develop their thought or bring more precision?

line 245: how different are these samples if we compare global gene expression data?

fig3B and 4E: AMP + Bom profile show dramatic difference: the authors should explain or at least mention and comment it in the text

line 279: there is only a minor difference, i am not sure it is significantly different (eater-/- data)

line 280: “indicates that…”: based on data, i am not sure I see that.

lines 282-283: it is too simplified, there are several differences between samples: d1 and d8 FB are different ; d1 and d8 Hemocyte are different etc. Could that be further described/analyzed?

lines 285-287: i am not sure this is true based on their data

line 308: figS5A: are OrR data the same as shown in FigS2E? If yes, please redo.

line 317: I somehow feel this is a weak ending. I see the value of that group of genes, but I feel somehow biological conclusions on the functional genetics that has a phenotype is more impressive.

Reviewer #3: Line 280: Figure 1F does not exist

PLOS authors have the option to publish the peer review history of their article (what does this mean?). If published, this will include your full peer review and any attached files.

Reviewer #1: No

Reviewer #2: No

Reviewer #3: No
---

## [Decision Letter · Decision Letter 1]

6 May 2023

Dear Dr. Wunderlich,

Thank you very much for submitting your manuscript "Drosophila immune priming to Enterococcus faecalis relies on immune tolerance rather than resistance" for consideration at PLOS Pathogens. As with all papers reviewed by the journal, your manuscript was reviewed by members of the editorial board and by several independent reviewers. In light of the reviews (below this email), we would like to invite the resubmission of a significantly-revised version that takes into account all the reviewers' comments.

Reviewer 1 and Reviewer 3 both remain dissatisfied with the statistical analyses.  They also share the concern about the confounding results with spz and myd88 mutants. Please be sure to address these concerns. 

We cannot make any decision about publication until we have seen the revised manuscript and your response to the reviewers' comments. Your revised manuscript is also likely to be sent to reviewers for further evaluation.

Sincerely,

Danielle A. Garsin

Guest Editor

PLOS Pathogens

Nina Salama

Section Editor

PLOS Pathogens

Kasturi Haldar

Editor-in-Chief

PLOS Pathogens

orcid.org/0000-0001-5065-158X

Michael Malim

Editor-in-Chief

PLOS Pathogens

orcid.org/0000-0002-7699-2064

Reviewer's Responses to Questions

**Part I - Summary**

Reviewer #1: In the revised manuscript entitled “Drosophila immune priming to enterococcus faecalis relies on immune tolerance rather than resistance” the authors have addressed some of the concerns brought up by the reviewers. It is not clear that the new statistical tests applied are more appropriate than the ones applied previously or that multiple comparisons have been taken into consideration. The addition of beads to the phagocytosis data provides additional confidence, but the spz data raise questions. As such, the data remain a collection of observations about the role of phagocytosis, IMD pathway, and spz in immune priming that do not clarify the process under study.

Reviewer #2: I feel that the authors have succesfully tackled my major comments

Reviewer #3: Cabrera et al. presents a characterization of the priming of Drosophila melanogaster innate immune system by Gram-positive E. faecalis. Adult animals subjected to a low dose of E. faecalis have improved survival against a second higher dose of the bacteria. The authors present evidence showing increased bacteria tolerance and phagocytic activity are key mechanisms of priming. The RNAseq analysis reveals interesting transcriptional changes involving metabolic shift associated with priming.

**Part II – Major Issues: Key Experiments Required for Acceptance**

Reviewer #1: 1) For all bacterial load experiments, the authors should consider applying their statistical test after log transformation of the data. This is a commonly used approach to skewed data and, in particular, bacterial load measurements that reduces the influence of outliers.

2) In Figure 1C and D, the investigators have moved to a hazard ratio instead of log rank analysis without providing justification other than that this is “more rigorous.” Has the author has modified the test for multiple comparisons as recommended by reviewer 3? In fact, although the survival traces are separated in Figures 1 C and D, since the authors acknowledge that these two panels utilize the same control traces (PBS/PBS and mock primed), a statistical test must take all four conditions (two controls, two tests) into account in the multiple comparison analysis regardless of how the data are presented.

3) Fig 3D and E and 4B: It is not clear to me that a pairwise statistical test can be applied to data extracted from an RNAseq experiment that involved multiple comparisons. While applying the appropriate test would not change the significance result, it is important to use statistics correctly.

4) The new data suggesting that Spz is required for priming but not Myd88 is confusing and requires more follow-up to be certain this is not an artifact of the mutant used, etc.

Reviewer #2: No experiment is required at this stage

Reviewer #3: 1. The authors’ previous analysis and conclusion implied Toll not being required for priming despite being essential for surviving the first infection. On the other hand, the Imd pathway was argued to be required for priming based. However, even with newly provided data the revised analysis does not seem to back up this claim, and frankly lacks statistical rigor.

Line 280-282: The authors claim that kenny, Tab2 and Relish mutants, similar to imd, have diminished immune priming. Upon closer inspection of the Efae Prime/Efae Mock in these data, only Rel[E20] seems to have a notably different priming Hazard Ration (HR=0.606) compared to OrR (HR=0.2855), albeit with a major caveat that the genetic backgrounds are not controlled. However, the Efae Primed/Efae Mock HR of imd (0.3942), kenny (0.2777) and Tab2 (0.3951) are very similar to Myd88 (0.3241), which the authors consider to have normal priming, and to OrR (wildtype). Just a curiously examination of these survival curvies, Figure S4, suggests priming is intact in most of the imd pathway mutants. Moreover, the comparison of Hazard Rations between these different mutants lacks rigorous statistical testing.

It seems that the authors still retained the conclusions drawn from the comparison of median survival time but have taken most of the sentences that mentioned this parameter out of the reasoning due to the lack of statistics. The authors should consider include median of survival time back (with statistics), or exclude it completely and draw new conclusion?

2. The lack of thorough consideration of and control for genetic background remains an issue. It is not at all clearer why OrR is an appropriate control for any of the mutants examined.

3. The differeing result with spz vs. MyD88 mutants are potentially important but could also be due to poorly control genetic backgrounds and requires more work to solidfy.

4. The connection by which tolerance, phagocytosis and metabolic shift all contribute to the mechanisms to priming is unclear, although this can be left for future investigation.

**Part III – Minor Issues: Editorial and Data Presentation Modifications**

Reviewer #1: (No Response)

Reviewer #2: (No Response)

Reviewer #3: - Supp Fig 1A: fly has gained re-infection survival from priming, some quantification would be appreciated here. HR between 1 day vs 7 days? Additionally

- P-values for the hazard ratios are present in supp table 1, these values should be included in the text and figures/figures’ legends.

- Why are eater flies surviving better in the PBS/PBS condition, compared to OrR flies?

- Is is possible to track the two subsets of low-dose injected flies (the resisting and the tolerating) to see which population survive better after the high-dose 2nd challenge? (Fluorescent microbes?)

- Line 324: Does “edges” mean “lines”?

- Line 372: “we were also identified as” – grammar mistake

- This reviewer appreciates the authors’ effort in explaining the network presentation of GSEA result. Perhaps the authors should highlight the significance of using this novel presentation method instead of using the traditional enrichment score plot?

PLOS authors have the option to publish the peer review history of their article (what does this mean?). If published, this will include your full peer review and any attached files.

Reviewer #1: No

Reviewer #2: No

Reviewer #3: No
---

## [Editor Report · Decision Letter 2]

19 Jul 2023

Dear Dr. Wunderlich,

We are pleased to inform you that your manuscript 'Drosophila immune priming to Enterococcus faecalis relies on immune tolerance rather than resistance' has been provisionally accepted for publication in PLOS Pathogens.

Best regards,

Danielle A. Garsin

Guest Editor

PLOS Pathogens

Nina Salama

Section Editor

PLOS Pathogens

Kasturi Haldar

Editor-in-Chief

PLOS Pathogens

orcid.org/0000-0001-5065-158X

Michael Malim

Editor-in-Chief

PLOS Pathogens

orcid.org/0000-0002-7699-2064
---

## [Editor Report · Acceptance letter]

7 Aug 2023

Dear Dr. Wunderlich,

We are delighted to inform you that your manuscript, "Drosophila immune priming to <i>Enterococcus faecalis<i/> relies on immune tolerance rather than resistance," has been formally accepted for publication in PLOS Pathogens.

Best regards,

Kasturi Haldar

Editor-in-Chief

PLOS Pathogens

orcid.org/0000-0001-5065-158X

Michael Malim

Editor-in-Chief

PLOS Pathogens

orcid.org/0000-0002-7699-2064